# Current state of mental health and substance use in Toledo, Belize: A mixed methods study

Jaclyn Kirsch[1] 📍, Megan Fabbri[2] 📍, Kristen Kerr[1] and Lynette Gomez[3]

[1]School of Social Work, University of Texas at Arlington, Arlington, TX, USA; [2]School of Social Work, West Virginia University, Morgantown, WV, USA and [3]Hillside Health Care, Eldridge, Toledo, Belize

## Research Article

mental health; substance use; Belize; mixed methods; Caribbean

**Corresponding author:**
Jaclyn Kirsch;
Email: jaclyn.kirsch@uta.edu

## Abstract

Mental health and substance use are increasingly pressing issues in communities across low-and-middle income countries, including Belize, particularly Toledo, the country's most rural and resource-limited district. Using community-based participatory research methods, this preliminary mixed methods study (quantitative n = 163; qualitative n = 10) aims to (1) investigate mental health symptoms and substance use patterns in a non-randomized sample of individuals from southern Belize and (2) explore community perspectives on mental health among community stakeholders. Findings show high levels of depression, anxiety, and suicidal ideation among survey participants, which were supported by qualitative interviews. While low levels of substance use were reported by survey participants, qualitative findings diverged and showed alcohol use to be of significant concern among participants. Our study highlights the critical need for increased research, advocacy, and policy implementation regarding mental health and substance use in Toledo and across Belize. Given the scarcity of mental health resources in Toledo, findings underscore the urgent need for policy interventions that expand access to psychiatric services, integrate community-based mental health approaches, and address socioeconomic drivers of poor mental health outcomes.

## Impact statement

This paper is one of the first known studies that reports on mental health and substance use among a sample of community members in the country of Belize. The findings of our study are imperative to increase understanding of mental health and substance use among this population and will assist in bringing awareness to the topic among stakeholders. The high levels of mental health symptoms among study participants show the critical need for increased support and understanding of mental health in the country. Locally, this study brings awareness to the topic of mental health and substance use and highlights the need for more research, advocacy, and policy on the topic.

## Introduction and background

Mental health continues to grow as a critical social work and public health concern around the world. Currently, 80% of the global mental health disease burden occurs in low- and middle-income countries (LMIC; Alloh et al., 2018). Alongside mental health concerns, substance use has emerged as a significant issue, often exacerbating existing mental health conditions (Nadkarni et al., 2023). While research and understanding of mental health and its treatment in low resourced areas has dramatically increased in recent decades (Sweetland et al., 2014; Keynejad et al., 2021), many communities continue to have limited capacity to screen, diagnose, and treat mental health and substance use concerns. In the Caribbean, where an understanding of mental health and substance use is limited, it is difficult to approximate the rates of mental health diagnoses despite the fact that mental health disorders account for an estimated quarter of the disease burden in the Latin America and Caribbean region (Pan American Health Organization, 2013). Tracking substance use in the region is also challenging, and precise rates are difficult to determine; however, the region has a long history of substance use deeply embedded in its culture, characterized by high levels of alcohol consumption, marijuana cultivation, and its role as a significant hub for drug trafficking (Reid et al., 2015). One such country is Belize, a small, diverse country in Central America. According to the Belize Ministry of Health and Wellness (2023), no epidemiological study on mental health or substance use has been conducted, posing a challenge in assessing the overall well-being of the population. Additionally, Toledo, the southernmost district of Belize, is the most rural and least resourced, which exacerbates challenges for communities in accessing mental health and substance use services. Given the limited understanding of mental health and substance use in Belize, coupled with the heightened vulnerabilities

faced by residents of Toledo, this study seeks to enhance knowledge of the current state of mental health and substance use within Toledo, Belize.

### Belize and Toledo

Belize is situated in Central America between Mexico, Guatemala, and Honduras. Despite having a population of fewer than 500,000 people, the country is remarkably diverse, with a wide array of ethnicities and languages (World Bank, 2022). The country's history of British colonialism has a profound impact on its socio-political and cultural landscape. The introduction of British colonial structures led to the displacement and marginalization of Indigenous Mayan communities, whose struggles for land rights and cultural recognition continue today (Shoman, 2010). Although Belize is classified as a middle-income country, rural areas face significant economic disparities with limited access to essential services like healthcare, education, and infrastructure (Government of Belize, 2010). The legacy of colonialism has contributed to structural inequalities, with indigenous and Afro-Belizean communities disproportionately affected by poverty, particularly in the Toledo district (Minority Rights Group, 2017). As the most rural district in Belize (Statistical Institute of Belize, 2023), Toledo is home to the highest percentage of Indigenous people, including the majority of the country's Mayan population—its poorest ethnic group (Gahman et al., 2020). The availability of social services, including access to mental health and substance use providers, throughout Toledo is notably limited, exacerbating the challenges faced by its residents (Belize Ministry of Economic Development, 2010).

### Mental health and substance use in Belize

The recognition and understanding of the need for mental health and substance use services in Belize have paralleled global trends over the past several decades. Before the 1990s, the country was not actively acknowledging the need for mental health care and continued the colonial era institutional models where those who were severely mentally ill were put into the single psychiatric hospital in the country and most others were left untreated (Killion and Cayetano, 2009). In 1991, the first training program was established for psychiatric nurse practitioners through the Pan American Health Organization (PAHO) to provide community-based care throughout the country (World Health Organization, 2009). While at the time this program was seen as incredibly successful in supporting the mental health of Belizeans, it currently is not meeting growing need as there are just 13 psychiatric nurse practitioners (PNP) working in the country through this program (Estephan et al., 2023). The persistent shortage of PNPs limits the availability and accessibility of services across the country, especially in rural areas like Toledo. This is exacerbated by insufficient training opportunities and resources for new PNPs, hindering the program's sustainability and ability to expand to meet current demands.

In Toledo specifically, the rural nature of the district, higher levels of poverty, and large Indigenous population lead to both increased mental health concerns and more limited access to structural supports for treatment options. This severely restricts access to care for residents of the district's 50+ villages, most of which are Indigenous communities with distinct conceptualizations of mental health that differ from traditional Western models, necessitating culturally specific treatment approaches (Hatala and Waldram, 2016). Many people living in Toledo, especially Indigenous Mayans, are more likely to seek treatment from traditional healers rather than providers in hospitals and clinics (Rich et al., 2022).

A key barrier to instilling appropriate mental health services in Belize is increasing the understanding of mental health concerns in the country. The recent National Mental Health Policy 2023–2028, released by the Belize Ministry of Health and Wellness (2023), provides information on mental health diagnostic categories from the Belize Health Information System (BHIS). The data reveal that among 11,312 recorded mental health visits from 2018 to 2021, the top three diagnoses were anxiety disorders (45%), affective disorders (22%), and schizophrenia and related disorders (8%). The report also showed that 32 people died by suicide (7.63/100,000 death rate) (Ministry of Health and Wellness, 2023). Within this report, no specific information was provided on the rate of those being treated for substance use disorders. In a study of patients from a primary care clinic in Belize City, researchers found that depression and anxiety were the most common diagnoses, with prevalence rates surpassing those reported in primary care clinics in other countries (Oladeji et al., 2024). No studies estimating the rates of substance use in Belize could be identified. The lack of comprehensive data on mental health and substance use in Belize hampers a full understanding of community needs, with particularly scarce information on the specific challenges faced in southern Belize.

The mental health treatment gap in Belize reflects broader trends in LMICs, where stigma, resource limitations, and workforce shortages hinder access to care (Evans-Lacko et al., 2018). This issue is particularly severe in rural areas like Toledo, where a shortage of mental health professionals and treatment facilities leaves many without adequate support (Nadkarni et al., 2023). Belize relies on a small number of psychiatric nurse practitioners, mirroring challenges in other under-resourced settings where community-based services remain underdeveloped (Keynejad et al., 2021). High levels of stigma further discourage individuals from seeking treatment, widening the gap between those in need and those who receive care. As a result, clinical samples fail to capture the true extent of mental health and substance use challenges in the broader population (Evans-Lacko et al., 2018).

### The current study

To address this research gap, our research team, in partnership with a local healthcare non-profit, sought to investigate rates of mental health and substance use among a population of individuals living in Toledo, Belize. Existing data on mental health and substance use in Toledo, Belize was limited and primarily drawn from clinical populations, leaving a gap in understanding among community-based individuals not actively engaged in treatment. To begin addressing this gap, this study aims to (1) examine the presence and patterns of mental health symptoms and substance use in a non-randomized community sample and (2) explore community perspectives on mental health through qualitative stakeholder interviews, providing critical insights into an understudied population. These preliminary findings contribute to the literature by providing insights into the current state of mental health and substance use in southern Belize, highlighting key areas for further research and intervention.

## Methods

### Design

This study was co-developed by the lead researcher and the Executive Director of a local non-profit healthcare organization in Toledo, Belize in response to the clinic's growing concerns about the mental health of its patients. The organization, a trusted provider in the community, offers free healthcare through its main clinic near Punta Gorda and a mobile clinic serving villages across the district. While it has no dedicated mental health practitioners, concerns are referred to the Ministry of Health. Recognizing the need for improved mental health support, the organization sought to better understand community needs and inform future efforts to enhance well-being.

We implored a community-based participatory research approach (CBPR) throughout the entirety of the project (Wallerstein et al., 2017), beginning with the establishment of an advisory committee that consisted of three staff members of the partner organization. This group met regularly throughout the project to assist in development of the study, recruitment of participants, data collection, and data analysis. Our study used an exploratory sequential mixed methods design (Creswell, 2014), where Phase I consisted of qualitative interviews with staff members of the partner organization (N = 10) that then directly informed the variables measured in Phase II where a quantitative survey was completed by community members (N = 163). A mixed methods approach was chosen to provide a more comprehensive understanding of mental health and substance use in Toledo as quantitative data allowed for the identification of symptom patterns, while qualitative insights captured community perspectives and contextual factors influencing mental health and substance use. This integration of methods ensured a deeper exploration of the topic beyond what either approach could achieve alone. The study was approved by the Institutional Review Board at the study site.

### Phase I: Qualitative

In April 2021, the first author conducted 10 virtual semi-structured interviews *via* Zoom due to COVID-19 travel restrictions. Interviews, lasting 45–90 minutes, were conducted in English with staff from the partner healthcare organization, including community health workers, clinic staff, and outreach coordinators. Staff members on the advisory committee were not eligible to complete an interview. Participants provided diverse perspectives on mental health and substance use, drawing from both their professional roles and lived experiences as community members of Toledo. Given the stigma surrounding mental health, staff members were selected for their existing relationship with the first author, ensuring a more open discussion. The virtual format limited opportunities to build trust with external community members, making staff the most suitable participants. This approach facilitated candid discussions while prioritizing participant comfort and confidentiality.

Participants completed a short demographic questionnaire and consented to participate in the study before each interview. Questions were developed in partnership with the advisory committee and aimed to uncover participants' viewpoints on mental health in the Toledo community, including perceptions of depression, anxiety, and substance use, as well as barriers to care, and the impact of the COVID-19 pandemic on well-being. Interviews were recorded, and transcriptions were generated using Zoom before being manually reviewed to ensure accuracy. Data collection ceased when saturation was met.

### Phase II: Quantitative

The quantitative survey was developed using both results of the qualitative interviews and input from the advisory committee to create and collect data that would be meaningful and useful in improving services. All scales were reviewed with the advisory committee to assess questions applicability to the Belizean cultural context. All questions and scales were found to be acceptable, and no changes were made. Survey data were collected *via* three staff members from the partner organization who were different than members of the advisory committee. The data collectors had no prior experience with data collection and were not trained mental health professions. Therefore, the research team provided training on data collection methods and ethical research practices, including best practices in collecting mental health data and handling sensitive topics. Training emphasized strategies for addressing potential concerns arising from asking community members stigmatized or personal questions about their mental health. They additionally completed the Collaborative Institutional Training Initiative (CITI) training on Human Subjects Research (HSR) to understand best practices in human subjects research. They used snowball sampling to recruit participants, a common approach when conducting research with understudied and hard-to-reach populations (Goodman, 2011).

Study participants in Phase II included any adult residing in the Toledo district, regardless of mental health history or healthcare engagement. This community-based sample was essential for capturing a broader understanding of mental well-being beyond clinical populations as previous research in Belize has primarily focused on individuals receiving treatment for mental health conditions. By including participants from the general community, this study provides a more comprehensive perspective on mental health and substance use patterns among the wider population. Because this study utilized non-randomized, snowball, and convenience sampling, a formal sample size calculation was not conducted as the recruitment process did not rely on probability-based sampling. Instead, participant recruitment was facilitated through community networks, allowing access to individuals who might otherwise be difficult to engage in mental health research. While this limits generalizability, it provides valuable insight into an understudied population and helps establish a foundation for future research. At the conclusion of the survey, participants were then asked if they knew others who may be interested in participating. Participants did not receive an incentive for their participation based on feedback from the advisory committee. Data collection concluded when data collectors reported they could no longer identify participants to complete the survey.

### Measures

The PHQ-9 and GAD-7, widely validated tools for assessing depression and anxiety (Kroenke et al., 2001; Spitzer et al., 2006), were used in this study. These measures have been applied in resource-limited settings, including the Caribbean (Carroll et al., 2020). Given the lack of prior validation in Belize, we report reliability statistics (PHQ-9: $\alpha = .84$; GAD-7: $\alpha = .88$) to assess scale performance in this context. All reported information on suicidal thoughts came from the final item of the PHQ-9, which asks participants whether they have had thoughts of self-harm or being better off dead in the past two weeks. Additionally, substance use was assessed using a self-reported frequency scale, where participants indicated their use of alcohol, cigarettes, and marijuana over the past month. Because this study aimed to capture general

substance use patterns rather than diagnose substance use disorders, a frequency-based measure was sufficient and allowed for a straightforward assessment without the need for a diagnostic, validated scale.

### Data analysis

Data for each phase was first analyzed separately before being integrated. We utilized thematic analysis, following Clarke and Braun (2017), with a deductive coding approach to examine qualitative data (Fereday and Muir-Cochrane, 2006). The data were analyzed and coded based on predefined mental health concerns, specifically focusing on depression and suicide, anxiety, and substance use. This method enabled us to systematically identify and interpret patterns related to these key issues within the participant responses. Multiple rounds of coding were completed by the second and third author based on the mental health concerns of interest. Initially, they familiarized themselves with the data by reading the transcripts multiple times. This immersion in the data facilitated a deep understanding of the context and nuances within the participants' narratives. They then applied the predefined codes to the data, categorizing segments of text that aligned with the mental health concerns under investigation. Throughout this process, they engaged in regular discussions with the first author and the advisory committee to resolve any discrepancies and refine the coding framework, ensuring that it accurately reflected the participants' experiences and perspectives. This collaborative approach helped us to enhance the rigor of the analysis as it minimized the potential for individual bias and ensured that the coding was both comprehensive and nuanced.

Quantitative survey data were analyzed using descriptive statistics, including frequencies and percentages in SPSS 27. Qualitative and quantitative results are reported together with themes being developed based on the distinct mental health outcomes measured in the quantitative survey, with the qualitative findings being integrated through a weaving technique, a process where quantitative and qualitative findings are reported on a concept-by-concept basis (Fetters et al., 2013). As a form of member checking, the advisory committee and two staff members who participated in the interviews reviewed and approved the results section to ensure accuracy and alignment with community perspectives.

### Results

Table 1 shows the demographic details of both the qualitative and quantitative samples. Interview participants mostly identified as female (80%), between 30 and 39 years old (60%), Ketchi (40%), and married (70%). A majority of survey participants identified as female (58.3%), between 25 and 29 years (37.4%), Ketchi (39.9%), and married (55.8%). Survey data were similar to data from the 2022 Belize Census for Toledo where just over half of the citizens are female (50.7%), around half are Ketchi (49.4%), and a little over half are married or partnered (56.4%).

### Depression and suicide

Table 2 shows the results from the PHQ-9. Quantitative data found that a majority of the survey respondents showed mild or no depressive symptoms (63.8%). However, 31.4% of respondents scored above the cut-off for depression, with 20.9% having moderate, 7.4% having moderately severe, and 3.1% having severe

**Table 1.** Demographic characteristics of study participants (*N* = 173)

| | Survey (*n* = 163)<br>*f (%)* | Interviews (*n* = 10)<br>*f (%)* |
|---|---|---|
| **Sex** | | |
| Male | 62 (38) | 2 (20) |
| Female | 95 (58.3) | 8 (80) |
| **Age** | | |
| 18–24 years | 31 (19) | 1 (10) |
| 25–29 years | 61 (37.4) | 2 (20) |
| 30–39 years | | 6 (60) |
| 40–54 years | 29 (17.8) | |
| 55–69 years | 23 (14.1) | 1 (10) |
| 70 and older | 16 (9.8) | |
| **Ethnicity** | | |
| Ketchi | 65 (39.9) | 4 (40) |
| Mopan | 13 (8) | |
| Garifuna | 18 (11) | 1 (10) |
| Creole | 10 (6.1) | |
| East Indian | 23 (14.1) | 3 (30) |
| Mestizo | 20 (12.3) | |
| Mennonite | 2 (1.2) | |
| Other | 3 (1.8) | 2 (20) |
| **Marital status** | | |
| Single | 46 (28.2) | 2 (20) |
| Married/partnered | 91 (55.8) | 7 (70) |
| Formally married/ partnered | 17 (10.4) | 1 (10) |
| **Occupation** | | |
| Support Staff | | 6 (60) |
| Clinical Staff | | 4 (40) |

depressive symptoms. Qualitative participants supported these findings, stating that they observed sadness and depressive symptoms among community members. One participant noted:

*I would say there is depression. Sometimes we require, you know, to visit them [patients] more often or have emotional support for them as well. Yeah, we also do hospice and palliative care like if a family member dies, then we follow up with them, because you know…that grieving stage that sometimes lead to depression as well* (10).

Others agreed saying, "*I think there is some sadness and depression in our community, for sure*" (2) and "*That's an issue in the community, people they don't really do anything about it*" (5). Participants noted that reasons for depression included financial concerns and family stress, with the two causes often overlapping. One participant stated: "*There is a lot of people that don't have jobs, they don't have as good of income to provide for the family all these stress, things like that, people get depressed*" (4). Another noted, "*I think I see some sadness and depression, especially from couples who was breaking up. And their families who have problems, like a mother and a father who has problems and are fighting*" (3).

**Table 2.** Depression and suicide (PHQ-9)

| | Not at all f (%) | Several days f (%) | More than half f (%) | Everyday f (%) |
|---|---|---|---|---|
| | 41 (25.2) | 49 (30.1) | 34 (20.9) | 5 (3.1) |
| Little interest or happiness in doing things | 57 (35) | 61 (37.4) | 26 (16) | 11 (6.7) |
| Feeling down, depressed, or hopeless | 48 (29.4) | 59 (36.2) | 31 (19) | 13 (8) |
| Trouble falling or staying asleep, or sleeping too much | 56 (34.4) | 58 (35.6) | 23 (14.1) | 16 (9.8) |
| Feeling tired or having little energy | 34 (20.9) | 66 (40.5) | 38 (23.3) | 13 (8) |
| Poor appetite or overeating | 60 (36.8) | 48 (29.4) | 26 (16) | 15 (9.2) |
| Feeling bad about yourself or that you have let yourself or your family down | 69 (42.3) | 45 (27.6) | 16 (9.8) | 25 (15.3) |
| Trouble concentrating on things, such as watching TV or YouTube | 89 (54.6) | 41 (25.2) | 19 (11.7) | 6 (3.7) |
| Moving or speaking slowly or quickly so that other people have noticed | 97 (59.5) | 40 (24.5) | 14 (8.6) | 3 (1.8) |
| Thoughts that you would be better off dead or of hurting yourself in some way | 101 (62) | 29 (17.8) | 10 (6.1) | 13 (8) |

**Table 3.** Anxiety (GAD-7)

| | Not at all f (%) | Several days f (%) | More than half f (%) | Nearly everyday f (%) |
|---|---|---|---|---|
| Feeling nervous, anxious, or on the edge | 50 (30.7) | 62 (38) | 33 (20.2) | 9 (5.5) |
| Not being able to stop or control worrying | 51 (31.3) | 60 (36.8) | 25 (15.3) | 18 (11) |
| Worrying too much about different things | 33 (20.2) | 62 (38) | 29 (17.8) | 33 (20.2) |
| Trouble relaxing | 66 (40.5) | 52 (31.9) | 25 (15.3) | 12 (7.4) |
| It is hard to sit still | 80 (49.1) | 38 (23.3) | 16 (9.8) | 15 (9.2) |
| Becoming easily annoyed or irritated | 49 (30.1) | 54 (33.1) | 29 (17.8) | 21 (12.9) |
| Feeling afraid as if something awful might happen | 46 (28.2) | 54 (33.1) | 28 (17.2) | 23 (14.1) |

In quantitative data, suicidal thoughts emerged as a concern. In the past two weeks, 62% of respondents reported having no thoughts of being "better off dead or of hurting yourself in some way." However, 17.8% indicated experiencing these thoughts several times a day, 6.1% reported such thoughts on more than half the days, and 8% experienced them daily. Most qualitative participants stated that suicide does happen, but it is rare. When asked directly, participants stated: *"I would say it's not really, it's not really at a high rate right now. But it does happen now and then."* (10), and *"No, it is not something we really see in the community"* (1). However, many participants noted anecdotal experiences of being told about suicidal thoughts from someone or seeing reports in the media. One participant noted that a family member who was experiencing domestic violence reported suicidal thoughts: *"There was once she said I wish I could just kill myself. Sometimes she thinks of just going and running in the bushes"* (3). Another stated, *"Well, in our village, we hardly hear that. We don't have people like that, but over the news, over the news we would hear that"* (6). Participants noted that the most common methods of suicide in the community were hanging and by ingesting poison. Participants also noted that those who did commit suicide tended to be men: *"Well, when it comes to suicide with women, women scarcely would do that [suicide]. It's mostly the men that commit suicide here in our country. I would say, like 90% it's men that commit suicide"* (9).

### Anxiety

Table 3 shows the results of the GAD-7. Quantitative and qualitative findings supported the notion that anxiety is a critical concern among community members. In the quantitative data, 36.2% of participants scored above the threshold for probable generalized anxiety disorder. Regarding anxiety severity, 29.4% of participants exhibited mild anxiety, 18.4% showed moderate anxiety, and 11% presented with severe anxiety symptoms. Qualitative findings revealed that participants strongly connected anxiety and stress to mental health. One participant stated:

> Mental health for me is basically my own thinking, it does not really have to be like some kind of psychosis but like I said earlier, it could be you know, anxiety, those, you know, those illnesses that are really not really easily expressed by people that are suffering from it (10).

Participants reported an increase in stress among community members, particularly after the COVID-19 pandemic, due to concerns about health and the financial impact on individuals. One participant stated:

> I think it's [COVID-19] impacted a lot of people, because a lot of people here they were working half of their job, half of their pay, water pay, some lost it and they were really stressed out. Everywhere you hear people talking like man this is one of the worst I've been (3).

Participants often noted the connection between stress and physical sickness, stating concern over community members' high levels of stress and the impact on health. One person noted, *"Yeah I cannot stress enough on that, because stress is the number one cause for hypertension and diabetes"* (4). When asked if finances were a cause of stress among community members, one participant stated:

> Oh that's, that everybody. That's everybody's stress here, but I do not know, right, but that's the biggest stress. Money, income and not only the money, the cost of living, the groceries in the store they went sky high, the price rise. You know in this hard time that's another stress because you are wondering how you could stretch or how you could divide that little money, to buy those things that went so high (9).

### Substance use

Findings regarding substance use (see Table 4) presented a notable divergence: while survey data reflected limited substance use overall, qualitative findings indicated high levels of substance use,

**Table 4.** Substance use

| | 0 day f (%) | 1 or 2 days f (%) | 3 to 5 days f (%) | 6 to 9 days f (%) | 10 to 19 days f (%) | 20 to 29 days f (%) | All 30 days f (%) |
|---|---|---|---|---|---|---|---|
| Cigarettes | 137 (84) | 8 (4.9) | 4 (2.5) | 3 (1.8) | 0 | 0 | 2 (1.2) |
| Cigars | 142 (87.1) | 2 (1.2) | 1 (.6) | 1 (.6) | 0 | 0 | 0 |
| Alcohol | 89 (54.6) | 39 (23.9) | 12 (7.4) | 5 (3.1) | 3 (1.8) | 2 (1.2) | 0 |
| Four (female)/Five (male) or more alcoholic drinks | 117 (71.8) | 20 (12.3) | 7 (4.3) | 1 (.6) | 4 (2.5) | 1 (.6) | 0 |
| Marijuana | 145 (89) | 1 (.6) | 0 | 1 (.6) | 1 (.6) | 0 | 3 (1.8) |
| Tobacco or chewing tobacco | 147 (90.2) | 1 (.6) | 0 | 0 | 0 | 0 | 0 |
| Other | 136 (83.4) | 1 (.6) | 0 | 0 | 0 | 0 | 0 |

particularly among men. A majority of survey participants stated that they had never used any substances. The most commonly reported substance used was alcohol (20.3%). Two participants (1.2%) stated smoking every day, three participants (1.8%) reported using marijuana every day, and none reported smoking cigars, consuming alcohol, using chewing tobacco, or using other drugs every day. About half of participants (54.6%) reported never consuming alcohol and a majority reported never heavily drinking (71.8%).

Interview participants stated high levels of concern for substance use in the community, mainly alcohol consumption among men. One participant stated: "*Substance abuse [is a key problem] because alcoholism, and you know in Belize people consume a lot of alcohol. Yeah I would say that is the biggest problem*" (2). An increase in drinking was additionally noticed during the COVID-19 pandemic where stress and anxiety led to increased levels of drinking:

> *Alcohol use is very high here, yeah, people like drinking. Well, again because of COVID people were like locked up most of them so, then they had to stay home, so a lot resorted to drinking and stuff like that well drinking was always a problem here, but to me it grew more* (1).

Finally, participants often discussed a connection between stress, alcohol consumption, and domestic violence. They reported a pattern where men were seen as stressed, which led to alcohol consumption, which led to domestic violence incidents between themselves and their partners. One participant stated: "*Yeah and then sometimes it [alcohol] brings a lot of problems, especially when it comes to going home, brings a lot of problems to homes, to the family. And so that is where the domestic violence come in*" (7).

### Discussion

Although earlier research has suggested potentially high rates of mental health issues among Belizeans (Ministry of Health and Wellness, 2023; Oladeji et al., 2024), this is the first study of its kind to investigate mental health and substance use rates among a group of Belizeans who are not currently undergoing treatment for mental health diagnoses. Our study found high rates of depressive symptoms, suicidal ideation, and anxiety symptoms through quantitative findings that were supported in qualitative interviews. Additionally, low rates of substance use were reported in quantitative findings, yet stated as a critical issue in qualitative interviews. Our findings are in line with those from other LMIC where mental health and substance use concerns are on the rise and of increasing concern for the overall well-being of community members (Alloh et al., 2018; Nadkarni et al., 2023). Our quantitative findings

showed similar or higher rates of depression and anxiety compared to many other LMIC countries. Among study participants, 34% screened positive for major depressive disorder, compared to 14% in Nepal (Kohrt et al., 2016), 11.5% in Ethiopia (Fekadu et al., 2017), and 9% in Mozambique (Cumbe et al., 2020). In terms of anxiety, 36% of study participants screened positive for generalized anxiety compared to 38% in Lebanon (Sawaya et al., 2016), 22.8% in Peru (Zhong et al., 2015), and 23.5% in China (Tong et al., 2016). Our findings are particularly concerning given that, using the same measures, the rates of depression and anxiety in general community samples are often higher compared to those reported in primary care settings, where mental health issues are typically reported at higher rates (Park and Zarate, 2019).

Belize faces numerous risk factors for elevated rates of mental health and substance use issues, but Toledo, as the most rural district with the highest poverty levels, presents particularly critical concerns (Statistical Institute of Belize, 2023). While Belize has worked hard to implement supports for mental health and substance use through its PNP program (Killion and Cayetano, 2009), our study shows the need for increased financial and human capital to better support the high mental health and substance use concerns of Belizeans, especially in Toledo. Interview participants pointed to financial concerns and family stress as leading causes of mental health challenges. Robust research exists showing the connection between those living in poverty and the detrimental impacts it can have on mental health outcomes (Ridley et al., 2020; Guan et al., 2022), often exacerbated by a lack of both economic and mental health resources (Bass, 2019). This is evident in Toledo, where the lack of social service infrastructure and governmental support for individuals in poverty and those with mental health concerns hinders Belizeans' ability to thrive and improve their lives.

Suicidal ideation emerged as a critical mental health concern with 31.9% of survey respondents reporting suicidal thoughts within the last two weeks and 8% reporting suicidal thoughts every day in the previous two weeks. Interview participants initially stated feeling like suicide was not a key concern, but many had anecdotal experiences of hearing about or knowing of a suicide that had occurred in the community. While our quantitative results are based on a single question and more in-depth research is needed to confirm these results, research in other Caribbean and Central American countries has found suicide to be a growing concern (Silverman et al., 2020), leading to the need for an increased understanding of suicide and suicidal ideation among this population.

Our study also offers essential insights into substance use in Toledo, emphasizing the importance of conducting further research to achieve a more comprehensive understanding of this issue. While survey participants reported low levels of substance use, qualitative interviews suggested significant concerns, particularly regarding alcohol consumption. This discrepancy likely stems from stigma and social desirability bias as individuals may have been hesitant to disclose personal substance use in surveys but were more willing to discuss it in relation to others. Similar patterns have been observed in other settings where stigma influences self-reporting on substance use (Yang et al., 2017). Given the association between alcohol use, masculinity, and socialization in Belize, future research should explore gendered norms and cultural perceptions of substance use. Incorporating alternative data collection methods, such as anonymous self-reports or biological markers, could provide a clearer picture of substance use patterns. Addressing stigma and improving community engagement will be essential in developing effective interventions.

## Limitations

This study provides critical baseline data on mental health and substance use in Belize, a valuable contribution given the scarcity of such research in the region. However, the study's limitations should be acknowledged. Firstly, the focus on participants from Toledo limits the generalizability of the findings to the broader Belizean population as well as to other LMIC and contexts. Future research should look to expand the study to more districts in Belize to provide a fuller understanding of mental health across the country. Additionally, the reliance on univariate quantitative data analysis restricts the ability to explore potential relationships between variables, which could offer deeper insights. The reliance on non-randomized, snowball sampling introduces potential bias, as recruitment was dependent on participant networks rather than probability-based methods. This may have led to overrepresentation of individuals with existing connections to the partner organization or greater trust in healthcare services, while potentially underrepresenting those disengaged from care or experiencing high levels of stigma. Future studies can mitigate potential bias by incorporating additional recruitment strategies, such as randomized community sampling or outreach through multiple, independent community-based organizations to capture a more diverse range of participants. The qualitative interviews, conducted in English rather than participants' primary languages, may have introduced communication barriers, potentially affecting the ability of participants to fully express their thoughts on the topic. Finally, our study was conducted in 2021 when COVID-19 lockdowns were still in place in Belize which may have impacted participants' mental health. These limitations suggest the need for future research with a more diverse sample, multivariate analyses, and culturally sensitive methodologies. However, even with these limitations, this study fills important gaps in the literature and helps to establish a baseline understanding of mental health and substance use in Toledo, Belize.

## Conclusion and implications

This study highlights the urgent need for increased attention to mental health and substance use concerns in Belize, particularly in rural and underserved areas like Toledo. While Belize has made commendable efforts to improve mental health services—such as the Belize National Mental Health Program, the Mind Health Connect initiative, and Galen University's recent establishment of a clinical social work program to train more social workers—significant gaps remain in service accessibility and utilization. A key policy implication of our findings is the need for greater financial and human capital investment to support the expansion of mental health services, ensuring that these efforts align with the National Mental Health Policy. Increasing the number of trained mental health professionals, particularly in rural districts, and integrating mental health services into primary care are critical steps toward improving access. However, even if services expand, deep-seated stigma surrounding mental health and therapy remains a significant barrier, making it difficult to engage individuals in treatment. Addressing this requires widespread mental health education and community outreach to normalize seeking care and reduce misconceptions about therapy.

Beyond healthcare, our findings underscore the broader socio-economic determinants of mental health in Belize. Poverty and financial stress emerged as major contributors to mental health struggles, highlighting the need for stronger governmental welfare support to improve economic conditions and provide social safety nets that can reduce financial strain on individuals and families. Without tackling these structural issues, mental health interventions alone may have limited impact. Further research is needed to investigate specific intervention strategies that can be culturally adapted and effectively implemented in Belize. While increasing access to therapy and psychiatric care is essential, future studies should also explore alternative community-based interventions that account for existing barriers such as stigma, financial hardship, and limited mental health literacy. Given the limited resources currently available, identifying cost-effective, scalable, and community-driven solutions will be key to improving mental health outcomes across the country. By addressing these policy gaps, strengthening economic support systems, and continuing efforts to reduce stigma, Belize can build on its existing progress to create a more comprehensive, accessible, and sustainable mental health care system for all its citizens.

**Open peer review.** To view the open peer review materials for this article, please visit http://doi.org/10.1017/gmh.2025.10007.

**Data availability statement.** The data that support the findings of this study are available from the corresponding author, Dr. Jaclyn Kirsch, upon reasonable request.

**Acknowledgements.** The authors would like to thank the staff and board of Hillside Healthcare International for their assistance and support throughout the project, specifically Alva Gomez, Amira Ack, Jennie Che, Mr. Matthew Nicasio and Dr. Carley Kirsch. Additionally, we thank Dr. Arati Maleku for her assistance and guidance throughout the research process.

**Author contribution.** Kirsch: conceptualization, funding, investigation, methodology, administration, supervision, validation, writing- original draft, writing-review & edits; Fabbri: formal analysis, validation, writing—review & edits; Kerr: visualization, writing—review & editing; Gomez: conceptualization, investigation, writing—review & editing.
*Contributions listed per CRediT role descriptors.

**Financial support.** This work was supported by The Ohio State University Office of International Affairs.

**Competing interests.** Authors report no conflicts of interest.

**Ethical statement.** This study was approved by the Institutional Review Board at the university within which the study was completed. All participants consented to both participate in the study and for their responses to be included

in published work. All data were securely stored on password-protected devices, and access was restricted to authorized research team members to maintain confidentiality.

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
