## [Reviewer Report]

The title accurately reflects the content of the paper, indicating a focus on mental health and substance use in Toledo, Belize, using a mixed methods approach. It is advised to consider the following suggestions:

1. Authors should succinctly summarize the summary, highlighting key findings and their implications, while clearly presenting goals, methods, and findings to engage the reader.

2. The introduction provides a good overview of mental health and substance use issues in low - and middle-income countries and the specific situation in Belize Toledo. However, the author should clarify the research gaps covered by the study and strengthen the links between global mental health issues and specific challenges faced in Belize, particularly in rural Toledo.

3. In the methodology section, the use of a community-based participatory research (CBPR) approach and an exploratory sequential hybrid approach to design is commendable. The author should elaborate on the rationale behind choosing a mixed methods approach and how it added value to the study compared to using a single method? The authors should provide more details on how the CBPR approach affects participant participation and data collection, and further elaborate on the limitations of snowball sampling and how to mitigate potential bias.

4. The results are clearly presented with tables supporting the text description. It is recommended to summarize the main findings in the text to supplement the table for readers who may not focus on the numerical data. Authors can also ensure a consistent integration of qualitative and quantitative research findings to provide a comprehensive understanding.

5. The discussion effectively linked the findings to broader issues in mental health and substance use in low - and middle-income countries. The authors should discuss the implications of the findings more broadly, especially in terms of policy and intervention strategies. In addition, authors need to compare the findings with other relevant studies to put their contributions in context.

6. The conclusions highlight the need for further research and policy development. It is recommended to strengthen the call to action by explicitly suggesting specific policy or program reforms, and to suggest avenues for future research that may build on the findings of this study.

7. The references are relevant and up-to-date, providing a strong foundation for the study. However, ensure all citations are properly formatted according to the journal’s guidelines.

---

## [Reviewer Report]

Current State of Mental Health and Substance Use in Toledo, Belize: A Mixed Methods Study

Abstract

This is a well written abstract.

Background/rationale

It is written well. However the section title is missing.

Methods

The methodology section is designed well and appropriate for the study questions and aims

Results

The results are presented clearly.

Discussion

The discussion relates well to the study aims, methods and findings.

Conclusion

This is a well written manuscript. It represents important findings on mental health and substance use patterns in Belize. It forms an important foundation for future studies.

---

## [Reviewer Report]

The manuscript addresses a substantial gap in mental health and substance use research in Belize. This research further focuses on the marginalized community in the Toledo district. The mixed-methods approach provides quantitative and qualitative insights, helping build a strong foundation for mental health data.

The manuscript’s exploration of depression, anxiety, suicidal ideation, and substance use provides a comprehensive understanding of mental health challenges in the community. The focus (impact statement) on the need for more research, advocacy, and policy implementation highlights the study’s contribution to global mental health.

Some sections (Methods and Discussion) include language that could be simplified for better readability. (i.e., the explanation of coding processes in the qualitative analysis may be condensed without losing detail).

The difference between the quantitative and qualitative findings regarding substance use is worth exploring further. Examining further how stigma might play a role in underreporting in the quantitative data could add depth to the findings and provide meaningful cultural insights.

Recommendations:

The abstract provides a comprehensive summary of the study. Including a sentence on the policy implications for Belize’s mental health infrastructure would further enhance its impact.

I would recommend adding more context about recent efforts to improve mental health services overall in Belize. While it’s clear that Toledo remains underserved, including an overview of the developments or progress that is happening could provide valuable context and help frame the challenges in the district more effectively (i.e., Belize National Mental Health Program (Ministry of Health and Wellness), Mind Health Connect, Galen University MSW program (clinical), School Counselor’s Association, etc).

The discussion section can be strengthened by referencing similar studies in neighboring LMICs or the broader Caribbean region to position the findings within a global context.

The manuscript highlights the need for further studies but could provide more specific recommendations, such as exploring, for example, the role of traditional healing practices in mental health interventions.

Overall, this manuscript makes a valuable contribution to the field of global mental health, particularly by providing baseline data for an underrepresented country of Belize.

---

## [Reviewer Report]

- you used n=163 for a survey and only 10 for interviews why the sample size is much different? If you want to use mixed methods you need a reliable sample size in both groups, provide your logic with reference.

The abstract and scaling procedure looks good.

- Please use the sampling design of the study instead of the design

- you used SPSS 27, please provide your data and code as a supplement file to reproduce the results

- Review the results again and correct if you find any inconsistencies

- and strength of your work.

- at the end of the discussion you need to provide some recommendations. The discussion is not well written overall. revise and improve it.

I made generic comments about the results. I Want to review it again after reproducing the results of getting data and code. Thank you.

---

## [Reviewer Report]

This is an essential work in this area and context. I would like to congratulate the authors for undertaking this critical study. The comments and reasons for the decision are provided in detail for the improvement.

Section 1: Introduction

•Line 12-15: The reference could be updated to a more recent one, rather than (Sweetland et al., 2014), which is nearly a decade old.

•The introduction could be more concise and focused on presenting a strong case for support. The excessive word count in the introduction might be better utilized in the methods or results sections.

Example: The information about Belize and Toledo could be more concise, and relevant references should be included. While this information is important for making a strong case for support, it could be condensed.

•Page 4-Line 19-21: It required a reference

•Repetition of information could be reduced

•Page 5: line “While these data are informative, they do not provide an accurate depiction of mental health and substance use across the entire population due to the treatment gap between those living with mental illness and substance use disorders and those who seek treatment in LMIC (Evans-Lacko et al. 2018)”: what are the factors for treatment gap in this context and is there any data discuss about the treatment gap in this particular context?

•Page 5: Line “This limited information regarding mental health and substance use in the country leads to limited understanding of needs.” At the same time, there were data documented in the Belize Health information system regarding the diagnosis of people who sought mental health care. This report included the rates of mental health conditions and mentioned substance use but not the percentage of SUDs. It is recommended to check the original data/survey results and record the reasons for non-documentation of substance use disorders.

Overall, the introduction could be more focused by reducing some of the information. It could begin by providing a broader perspective, then narrow down to the mental health prevalence and challenges within the Belize context. The introduction should also highlight the literature gap, the need for this specific study, and how the study will address that gap. The background information should effectively build a strong case for supporting the study.

Section 2. Methods

Design

•The argument for the community participatory approach concept is unclear as the participatory approach in research has different dimensions such as engagement, involvement and participation with the stakeholder. The current explanation leads to collaboration rather than using a community participatory approach.

Example: Where the stakeholders other than collaborating organisations involved in planning and co-designing? If that includes only data collection from the community samples, then that would be just a ‘participation’ in the research, which is typical for all the research with the human population.

•The reason for choosing only the staff of the organisation for the qualitative interviews needs to be justified considering the research question as it explores the mental health and substance use-related issues of the community participants. The limitation of this approach would be that it only considers the ‘Patient Reported Outcome Measures’ and omits ‘Patient Experienced Outcome Measures’ along with it, however, attempting a data triangulation!

•The sample size calculation for the quantitative phase was not clearly specified, which further limits the generalizability of the findings. The question of whether these samples accurately represent the population remains uncertain.

Example: Do these findings represent the particular community population? Since the study is a kind of prevalence type, a small number of samples may overrepresent the findings.

•Information regarding ethical approval needs further clarification, as it is important to understand details about the ethical review board and how data management and protection were handled, especially since the study used both online and face-to-face data collection methods.

•A brief about the partnering organisation (in one or two sentences) in the background would be useful for the readers to get an idea well and necessary as they were involved throughout the research and recruited samples for the qualitative phase from there.

•Qualitative: The inclusion and exclusion criteria have not been mentioned in the qualitative phase, including who this staff was, their background, and their experience, as they were asked for information related to their observations of mental health issues. If the staff are mental health professionals, they would have a better idea than someone from non-mental health or admin staff!

•Further, who collected data, whether they were trained, the medium of interviews, why an online platform etc, is essential information in this section and sample size.

•Quantitative: “All scales were reviewed with community partners to assess questions' applicability to the Belizean cultural context. No changes were suggested, and all scales and questions were acceptable.” This was unclear and convincing during the process. Who were these partners other than the one collaborating organisation?

•The staff were trained to collect data and this is a good practice. However, it is essential to mention the mental health background of the staff who collected the data as they are assessing depression and anxiety among the participants.

•“Survey data was collected via three staff members from the partner healthcare clinic.” This again indicates more information about the partner organisation as the readers do not understand the settings, which would limit them from understanding the study process.

•The study employed purposeful/convenience sampling along with the snowball sampling technique. Why is snowball sampling being used as a sampling method? Is the sample drawn from the community or specifically from individuals with various mental health issues? This aspect needs clarification, as it highlights the importance of having clear inclusion and exclusion criteria for participants or defining who the participants were in this phase.

•Information on PHQ 9 and GAD 7: the information could be reduced by giving the reason for choosing this scale and citing a reference.

•The reason for choosing the assessment tool for substance use is not clear

Data analysis

• Qualitative analysis: The framework used for the qualitative analysis is suitable and appropriate

• Quantitative analysis is key, and the analysis was not clearly mentioned (for example: Is it descriptive by using mean and SD or frequency and percentage)

• Triangulation of data: The study collected both qualitative and quantitative data and missing information related to the plans for the triangulation of data.

Section 3. Results

Quantitative:

1. The socio-demographic details need to be more elaborated, including information on socioeconomic status, education, employment, and other relevant factors. Since the study aims to assess mental health and substance use issues among participants from a rural community in Belize, these details are essential.

2. Denominator is important, out of how many of these findings? Breaking of findings in each of these conditions based on gender and other socio-economic variables is highly recommended to get more clarity

3. Data related to suicidality/suicidal ideas: PHQ 9 scale’s 9th question is related to suicidality. Do the authors use any other scales?

4. The association between socio-demographics and prevalence needs to be explored.

5. The triangulation of qualitative and quantitative needed to improve further,

Example: Qualitative finding ““There is a lot of people that don’t have jobs, they don’t have as good of income to provide for the family all these stress, things like that, people get depressed” needed to be triangulated with quantitative socio-demographic findings and that information was not mentioned in the results.

Qualitative

The facilitators and barriers to mental healthcare have not been reported in the results and this was one of the objectives of the qualitative interview with the staff.

The significant drawbacks of the findings are,

1. The fundamental question of whether these samples represent the larger community as a sample calculation has not been done.

2. Why use a mixed method instead of survey methods and only common mental disorders and substance use?

3. Patient Reported outcome measures and staff experience outcome measures! Instead, the generalisability would have been more accurate if the qualitative data had been collected from the same participants along with the quantitative data and staff could be additional to ensure the quality of findings.

Section 4. Discussion

• Discussion needed to be focused on the key findings.

Section 5. References

• Online/website information required date of access

• Incomplete references

Example: Bolland ON (2003) no place of publication and chapter or page!

Please check the submission guidelines for the references

---

## [Editor Report]

Thank you for submitting your manuscript to Global Mental Health. We have received comments from 5 reviewers who all provided similar feedback. As you will note in their comments, there is some information missing from the manuscript, particularly in the methods section, that we request the authors add to improve clarity and reproducibility. The reviewers have also provided some suggestions for improving the analysis and results, including the triangulation of qualitative and quantitative information. We hope you find these comments useful and consider revising your manuscript accordingly.

---

## [Reviewer Report]

Dear Editor,

I greatly appreciate the opportunity to evaluate this revised manuscript. I have thoroughly examined the updated manuscript, which incorporates the author(s) responses and modifications to the document. I have no further revisions.

Thanks.

Kind regards,

---

## [Reviewer Report]

The revised manuscript has effectively incorporated all previous recommendations, resulting in a clearer, more comprehensive, and well-contextualized study on mental health and substance use in Belize. The improvements in readability, particularly in the Methods and Discussion sections, enhance accessibility without compromising detail. The refined explanation of the qualitative coding process is now more concise, and the discussion of stigma’s role in substance use reporting adds important cultural nuance.

Notably, the manuscript now provides a stronger policy connection, with the abstract explicitly addressing the implications for Belize’s mental health infrastructure. The inclusion of recent mental health initiatives, such as the Belize National Mental Health Program and Mind Health Connect, strengthens the discussion by situating the study within ongoing national efforts. Additionally, linking the findings to broader global and regional research enhances the study’s relevance beyond the Belizean context.

The manuscript also now offers more concrete recommendations for future research, including the exploration of traditional healing practices in mental health interventions. These additions further contribute to the study’s impact on both research and policy.

Given these thoughtful revisions and strengthened contributions, I recommend acceptance of the manuscript. Congratulations to the authors on this important and impactful work.

---

## [Editor Report]

Congratulations to the authors on an excellent revision of the manuscript. Both the reviewers and I appreciate the thoughtful and substantial improvements made, and we are pleased to recommend the manuscript for publication.